

# Unveiling water quality and health risks from groundwater chemicals in Poyang Lake basin of China: a sophisticated analysis

Xiaodong Chu[1], Jingyuan He[1], Ting Chen[1], Hailin You[2,3], Xuhui Luo[4], Shuping Liu[4], Jinying Xu[2,4] and Zhifei Ma[4]

[1] Groundwater and Wetland Environment Research Laboratory, Jiangxi Provincial Coal Geological Exploration Research Institute, Nanchang, China
[2] Poyang Lake Wetland Observation and Research Station, Nanjing Institute of Geography and Limnology, Chinese Academy of Sciences, Nanchang, China
[3] Institute of Microbiology, Jiangxi Academy of Sciences, Nanchang, China
[4] Key Laboratory of Poyang Lake Environment and Resource Utilization, Ministry of Education, School of Resources and Environment, Nanchang University, Nanchang, China

Corresponding authors
Jinying Xu, xujy2020@ncu.edu.cn
Zhifei Ma, zfma919@126.com

## ABSTRACT

**Background**. Groundwater is a critical water resource in the Poyang Lake basin, especially given the increasing frequency of extreme drought events. However, comprehensive assessments of its chemical characteristics and associated health risks remain insufficient. This study aims to provide a comprehensive assessment of groundwater quality and associated health risks in the Poyang Lake basin, China.

**Methods**. This study collected 670 groundwater samples from domestic and agricultural wells across the basin during the 2022 dry season. Hydrochemical parameters, including manganese (Mn), ammonia nitrogen ($NH_{4+}$), iron ($Fe^{3+}$), aluminum ($Al^{3+}$), chemical oxygen demand (COD), fluoride ($F^-$), and nitrate ($NO_{3-}$), were analyzed and compared against China's Class III groundwater quality standards. Spatial variations in water quality were evaluated using the Entropically Weighted Water Quality Index (EWQI). Non-carcinogenic health risks via ingestion and dermal contact pathways were quantified through deterministic risk assessment models, with uncertainty and probability analyzed *via* Monte Carlo simulations (10,000 iterations). Sensitivity analysis was conducted to identify dominant factors influencing risk outcomes, ensuring a robust probabilistic interpretation of health hazards.

**Results**. The results unveils pronounced spatial variability across key hydrochemical parameters: pH ranges from 3.05 to 11.09, total dissolved solids (TDS) from 25.38 to 1,635.21 mg/L, Mn from below detection limits to 19.93 mg/L, COD from below detection limits to 8.57 mg/L, and TH from 9.47 to 905.78 mg/L. A notable proportion of samples breach the Class III groundwater quality standards, with 38% for Mn, 39% for COD, and 15% for $NH_{4+}$. The Piper diagram analysis categorizes the primary groundwater type as $HCO_{3-}$ -Ca, with a secondary Cl-Ca-Mg type. Through the EWQI assessment, while the overall water quality is deemed acceptable, we identify 56 sites with EWQI values indicative of poor to very poor quality, predominantly influenced by Mn and $NH_{4+}$. The irrigation risk assessment highlights extensive areas unsuitable for irrigation, particularly within the Ganjiang River basin. Health risk assessments utilizing a deterministic model reveal significant non-carcinogenic risks from $F^-$ and

$NO_3^-$ in groundwater, especially for children. However, our Monte Carlo simulation indicates that the probabilities of $F^-$ and $NO_3^-$ posing non-carcinogenic health risks are virtually zero, suggesting that the deterministic assessment may have overestimated the health risks. This study provides critical insights into the groundwater quality and health risks in the Poyang Lake basin and underscores the necessity for targeted water management strategies to mitigate pollution sources and safeguard public health.

## INTRODUCTION

Water scarcity poses a global crisis, affecting 4 billion individuals through periodic shortages and leaving over 2 billion without adequate water access (*Rahaman et al., 2024*). In developing countries, where water quality surveillance is lax, water-borne diseases account for 80% of illnesses, underscoring the critical need for improved water quality control. Groundwater serves as a vital source of water for human populations and ecosystems worldwide, underpinning agricultural, industrial, and domestic water supplies (*Bostic et al., 2023*; *Karandish, Liu & De Graaf, 2025*). Its reliability and quality are crucial for sustaining life and ensuring social development. However, the increasing threat of groundwater contamination, driven by human activities and climate change, has elevated concerns over its safety and sustainability (*Chakraborty et al., 2020*; *Kumar et al., 2024*; *Mukherjee et al., 2024*). The contamination of groundwater with arsenic (As) is particularly alarming, as it poses severe health risks, including increased mortality rates due to cardiovascular diseases and various types of cancer (*Chakraborty et al., 2020*; *Mao et al., 2023*; *Soldatova et al., 2022*). Nitrate pollution in groundwater can lead to serious health problems, such as methemoglobinemia in infants, and have negative environmental impacts, including eutrophication of surface water and global climate change (*Mao et al., 2023*). This contamination can disrupt the balance of ecosystems and is a major concern for public health, highlighting the urgency to assess and mitigate the risks associated with As contamination in groundwater.

Previous studies have primarily concentrated on using deterministic health risk assessment models to evaluate health risks in groundwater, based on the guidelines of the United States Environmental Protection Agency (USEPA) (*USEPA, 2011*). These models typically assume a certain value for each parameter, neglecting the uncertainties arising from sample point distribution, medium heterogeneity, testing errors, and human factors. This can lead to significant uncertainties in the results (overestimating or underestimating the actual risk) (*Gao et al., 2022*; *Mishra et al., 2021*; *Yuan, Li & Guo, 2023*). Fortunately, these issues can be addressed by employing probabilistic approaches, such as Monte Carlo simulations. Monte Carlo simulations utilize random sampling and statistical testing methods to generate random data and obtain approximate model solutions (*Gao et al., 2022*). The advantage of these simulations is that they can incorporate inherent variability and uncertainty in risk assessment, providing a more meaningful estimate of the probable

range rather than point estimates. Thus, they are effective in addressing model uncertainty. In this study, we combined deterministic models with Monte Carlo simulations to conduct a more comprehensive assessment of health risks in groundwater.

Poyang Lake, the largest freshwater lake in China, boasts abundant groundwater resources that are vital for maintaining regional ecological balance and supporting economic development (*Xu et al., 2021*). However, extreme climate changes in recent years have led to frequent droughts in the Poyang Lake basin, even during the flood season (*Liu et al., 2020*; *Peng et al., 2024*). Such circumstances have rendered the evaluation of groundwater health an urgent priority. Prior investigations into Poyang Lake's groundwater have delved into its chemical attributes and pollution origins (*Chu et al., 2022*; *Soldatova et al., 2022*). These studies, however, have predominantly zeroed in on particular regions or singular contaminants, falling short of a holistic assessment of the groundwater's overall health across the entire basin. Besides, few study combine deterministic risk assessments and Monte Carlo simulations for a comprehensive evaluation of both certain and uncertain health risks in Poyang Lake drainage.

Incorporating the insights from the preliminary analysis, this study focuses on several key research areas: (1) conducting a thorough assessment of the groundwater quality in the Poyang Lake basin; (2) evaluating the potential risks for irrigation and the carcinogenic and non-carcinogenic health risks linked to the consumption of groundwater in the region; (3) performing a sensitivity and uncertainty analysis on the health implications of groundwater use in the basin. The aim is to provide a comprehensive appreciation of the groundwater quality and the associated health risks in the Poyang Lake basin, offering invaluable data to inform decisions and guide policies for water resource management and public health.

## MATERIALS AND METHODS

### Study area

Poyang Lake, as China's largest freshwater body, is strategically located in the northern Jiangxi Province, within the subtropical monsoon climate zone of the Yangtze River basin (Fig. 1). The region experiences mild temperatures with a mean annual temperature of 16.7 °C and significant seasonal precipitation, averaging 1,400 to 2,400 millimeters annually and peaking in the summer, which significantly contributes to the lake's hydrological resources (*Liu et al., 2025*; *Xu et al., 2022*). The lake's water sources are predominantly fed by five major rivers—Ganjiang, Fuhe, Xinjiang, Raohe, and Xiushui—which bring vital water and nutrient loads to the ecosystem. The diversity of groundwater within the basin is marked by various types, including unconsolidated rock pore water, carbonate rock fracture-cave water, red clastic rock pore-solution fracture water, and bedrock fracture water (*Mao et al., 2021*). In recent years, human activities have significantly influenced the groundwater chemistry of the Poyang Lake Basin. For instance, agricultural activities in the Ganjiang region have led to increased nitrate levels in groundwater (*Wu et al., 2022*). In addition, the increase in extreme drought events has likely reduced groundwater recharge and caused water levels to drop (*Liu et al., 2025*). During the dry season, more groundwater flows into surface water, thereby altering the groundwater's chemical characteristics (*Fan et al., 2024*).

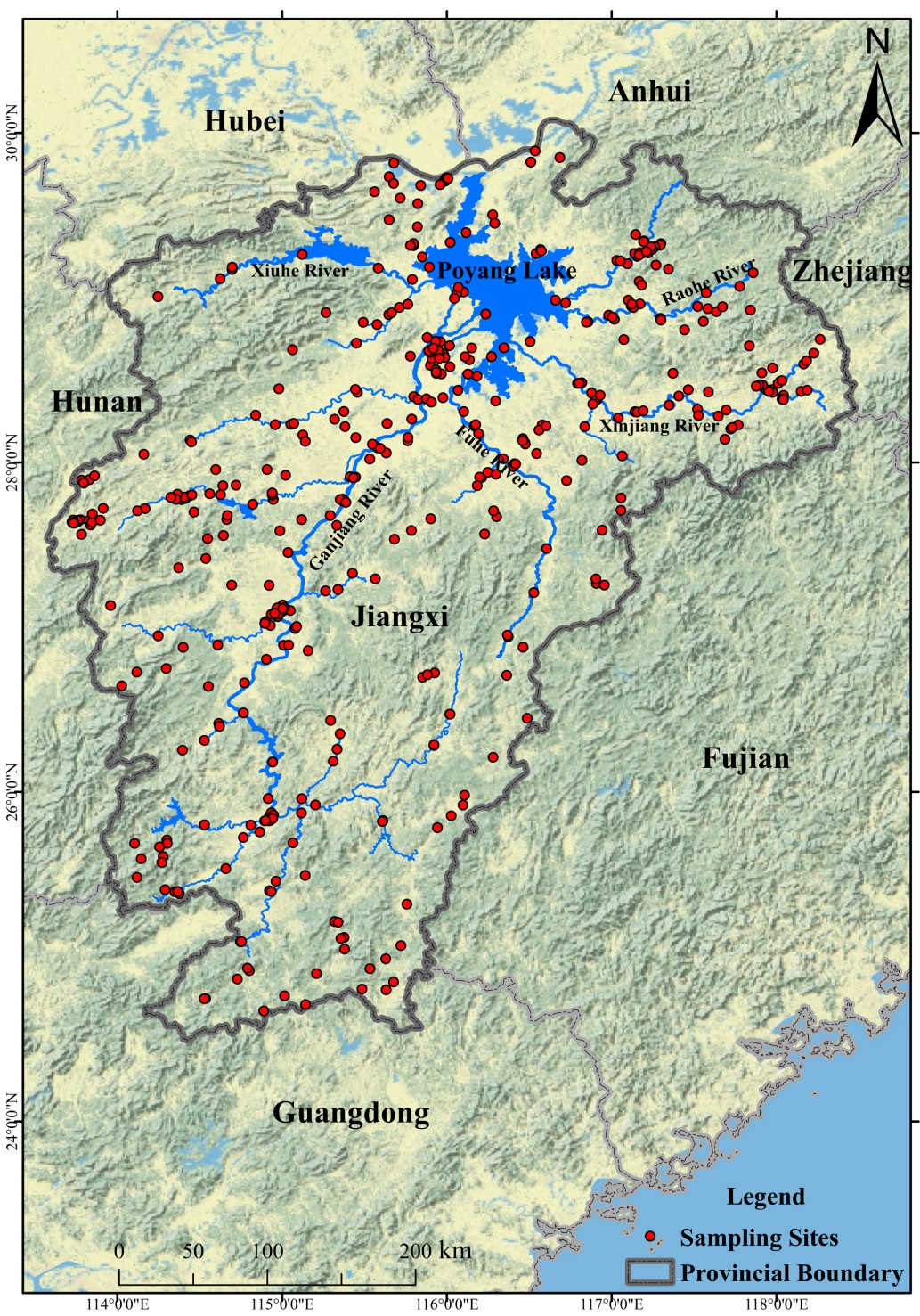

**Figure 1** Study area and sampling sites.

## Field sampling and analysis

During the dry season (specifically from December to March) in 2022, an extensive groundwater sampling campaign was executed, collecting 670 sampling points from rural domestic wells across the Poyang Lake basin. This initiative was conducted in strict accordance with the protocols established by the State Environmental Protection Administration in 2004. The samples were meticulously collected in polyethylene bottles that had been pre-cleaned, following a 10–15 min purge of the wells to ensure representativeness. The parameters including sodium ($Na^+$), fluoride ($F^-$), nitrate ($NO_3^-$), sulfate ($SO_4^{2-}$), manganese (Mn), biochemical oxygen demand (BOD), ammonia nitrogen ($NH_4^+$), total dissolved solids (TDS), total hardness (TH), pH, iron ($Fe^{3+}$), aluminum ($Al^{3+}$), and chloride ($Cl^-$) were chosen based on the standard for groundwater quality (GB/T 14848-2017). A liter of each groundwater sample was treated with concentrated nitric acid to maintain a pH of 2, thereby stabilizing the samples for subsequent analysis. Field measurements of pH were meticulously recorded using portable pH meters, ensuring the integrity of the data. The analytical procedures adhered to standardized methods, with rigorous quality assurance (QA) and quality control (QC) measures in place. These included the collection of one replicate sample for every ten samples as part of the field sampling process, followed by triplicate analyses to ensure the precision of the results. The limit of detection (LOD) for each parameter was determined based on the standard deviation of 20 reagent blank samples, multiplied by three.

## Water quality assessment

### EWQI assessment

Entropically Weighted Water Quality Index (EWQI) method is a technique for analyzing the distribution of groundwater quality in a study area. The approach involves determining the weight of water quality indicators through information entropy and condensing a large amount of water quality data into a representative numerical value that reflects the water quality condition (*Nisar et al., 2024*). Based on the calculated EWQI values, the groundwater quality in the study area can be divided into five levels from ''very poor'' to ''excellent'' as shown in Table 1. The calculation process for the Entropically Weighted Water Quality Index (EWQI) is intricate and involves five key steps: (1) constructing the initial water quality matrix (Eq. (1)); (2) standardizing the data (Eqs. (2) and (3)); (3) determining the weights (Eqs.(4) and (5)); (4) setting the quantification criteria for classification (Eq. (6)); and (5) calculating the EWQI itself (Eq. (7)).

$$X = \begin{pmatrix} x_{11} & x_{12} & \cdots & x_{1n} \\ x_{21} & x_{22} & \cdots & x_{2n} \\ \vdots & \vdots & \vdots & \vdots \\ x_{m1} & x_{m2} & \cdots & x_{mn} \end{pmatrix} \tag{1}$$

$$y_{ij} = \frac{x_{ij} - \min(x_j)}{\max(x_j) - \min(x_j)} \in (0, 1) \tag{2}$$

**Table 1 The criteria for EWQI and irrigation risk assessment.**

| Parameters | Range | Classification | Parameters | Range | Classification |
|---|---|---|---|---|---|
| EWQI | <25 | Excellent | MHR | <50 | Suitable |
| | 25–50 | Good | | >50 | Unsuitable |
| | 50–100 | Medium | PI | <80 | Good |
| | 100–150 | Poor | | 80–120 | Moderate |
| | ≥150 | Very Poor | | 120–200 | Poor |
| SAR | 0–10 | Excellent | SSP | <20 | Excellent |
| | 10–18 | Good | | 20–40 | Good |
| | 18–26 | Permissible | | 40–60 | Permissible |
| | >26 | Doubtful | | 60–80 | Doubtful |
| KR | <1 | Suitable | | | |
| | >1 | Unsuitable | | | |

$$p_{ij} = \frac{1+y_{ij}}{\sum_{i=1}^{m}(1+y_{ij})} \in (0,1) \tag{3}$$

$$e_j = -\frac{1}{In(m)}\sum_{i=1}^{m} p_{ij} In(p_{ij}) \tag{4}$$

$$w_j = \frac{1-e_j}{\sum_{j=1}^{n} 1-e_j} \in (0,1) \tag{5}$$

$$q_{ij} = \begin{cases} (c_{ij}/s_j) \times 100 \\ |(c_{ipH}-7)/(8.5-7) \times 100| \end{cases} \tag{6}$$

$$EWQI_i = \sum_{j=1}^{n} w_j q_{ij} \tag{7}$$

where, X is the initial data matrix with m groundwater samples and n water–quality indicators, and $x_{ij}$ is the original value of the jth indicator in the ith sample. $\max(x_j)$ and $\min(x_j)$ are the max and min values of the jth indicator. $e_j$ isthe information entropy of the jth indicator, $w_j$ the entropy weight, with lower $e_j$ meaning greater indicator impact. $s_j$ is the standard allowable value for the jth indicator (except for pH) in the Chinese groundwater quality standard, in mg/L. $c_{ij}$ is the standard allowable value for the jth indicator (except pH) from China's groundwater quality standard, in mg/L. The weight of chemical characteristics for computing EWQI is shown as Table S1.

*Irrigation risk assessment*

Groundwater serves as a crucial source of irrigation water in the Poyang Lake basin during dry seasons or periods of extreme drought. Given the significant impact of irrigation water quality on crops, it is essential to evaluate the water quality from the perspective of irrigation feasibility. In this study, the quality of irrigation water is assessed using indicators such as the sodium adsorption ratio (SAR), soluble sodium percentage (SP), Kelley's ratio (KR), permeability index (PI), and magnesium hazard ratio (MHR).

**Sodium adsorption ratio**: this index evaluates the propensity of sodium ions to accumulate in the soil and plants due to irrigation, which can affect soil structure and plant growth. The formula for calculating SAR is given by:

$$SAR = \frac{Na^+}{\sqrt{(Ca^{2+} + Mg^{2+})/2}}. \tag{8}$$

**Soluble sodium percentage (SSP)**: This percentage indicates the level of sodium ions that are soluble and potentially harmful to the soil. The calculation for SSP is as follows:

$$SSP = \left(\frac{Na^+ + K^+}{Na^+ + K^+ + Ca^{2+} + Mg^{2+}}\right) \times 100. \tag{9}$$

**Magnesium hazard ratio**: MHR indicates the potential risk that magnesium ions pose to crops, which can be detrimental in high concentrations. The MHR is determined by:

$$MHR = \frac{Mg^{2+}}{Ca^{2+} + Mg^{2+}}. \tag{10}$$

**The permeability index (PI)**: an indicator that measures the proportion of sodium ions relative to calcium and magnesium ions in irrigation water to assess the potential impact of the water on soil structure and permeability.

$$PI = \frac{Na^+ + \sqrt{HCO_3^-}}{(Na^+ + Ca^{2+} Mg^{2+})} \times 100. \tag{11}$$

**The Kelley ratio (KR)**: an indicator that measures the proportion of sodium ions relative to calcium and magnesium ions in irrigation water to assess the potential impact of the water on soil structure and permeability.

$$KR = \frac{Na^+}{Ca^{2+} + Mg^{2+}}. \tag{12}$$

The assessment criteria of these indicators are shown in Table 1.

## Health risk assessment
### Deterministic model

Following the 2011 guidelines of the United States Environmental Protection Agency (USEPA), deterministic models are commonly used to assess potential health risks. These models assign specific values to parameters in the formulas, reflecting group characteristics, to evaluate the health impacts of individual groundwater samples. GIS technology, such as Kriging interpolation, is utilized to map the spatial distribution of these risks. Health

risk assessments typically include evaluations of cancer risks (CR) and non-cancer hazard indices (NCHQ), with exposure mainly occurring through ingestion *via* drinking water and dermal contact during bathing. In this study, risks were assessed by calculating the chronic daily intake (CDI) for oral exposure and the dermal absorbed dose (DAD) for skin exposure, aggregating CR and NCHQ values from different exposure pathways for contaminants in the groundwater. The acceptable threshold for CR is set at $1 \times 10^{-4}$, indicating an acceptable lifetime cancer risk below this level. Groundwater is considered a health risk and unsafe if the NCHQ exceeds 1. Detailed information for the computing process is shown as following:

$$CDI = \frac{CW \times IR \times EF \times ED}{BW \times AT} \tag{13}$$

$$DAD = \frac{DA_{event} \times EV \times ED \times EF \times SA}{BW \times AT} \tag{14}$$

$$DA_{event} = K_P \times CW \times t_{event} \times 10^{-3} \tag{15}$$

$$NCHQ_{Oral-i} = CDI_i / RfD_i \tag{16}$$

$$NCHQ_{Dermal-i} = DAD_i / RfD_i \tag{17}$$

$$NCHQ_i = NCHQ_{Oral-i} + NCHQ_{Dermal-i} \tag{18}$$

$$NCHQ_{total} = \sum_{i=1}^{4} NCHQ_i = NCHQ_{F^-} + NCHQ_{NO_3^-} \tag{19}$$

where, CW represents the contaminant concentration in groundwater, expressed in mg/L. IR refers to the ingestion rate, in L/d, while EF signifies exposure frequency, in d/a. ED indicates the exposure duration in years to a specific chemical through drinking water consumption. BW is the average body weight in kg, and AT stands for the average exposure time in days. For dermal exposure, DAevent is the absorbed dose per event, in $mg/cm^2$. SA represents the skin area exposed to contaminants, measured in $cm^2$, and EV denotes the event frequency. $K_p$ is the permeability coefficient for a substance through skin in water, given in cm/h. $t_{event}$ refers to the duration of a single contact event in hours. RfD is the benchmark dose, in mg/kg d, with specific values for $F^-$ and $NO_3^-$ being 0.04 and 1.6 mg/kg d, respectively. The parameters for the calculations are listed in Table S2.

*Uncertainty analysis*

The Monte Carlo model is a computer simulation technique that utilizes random sampling and statistical methods to mimic the behavior and outcomes of complex systems. In health risk assessments, it accounts for uncertainties and variability in input parameters such as pollutant concentrations, ingestion rates, body weight, and exposure frequency. The process involves establishing probability distributions for these inputs, randomly selecting values that reflect the diversity of the population's characteristics, and then conducting numerous iterations—like 10,000—to generate a distribution of risk outcomes. A key advantage of the Monte Carlo model is its ability to reflect the inherent variability and uncertainty in input parameters, offering a range of risk estimates rather than a single value. This makes it particularly effective for managing uncertainties stemming from sample distribution, medium heterogeneity, measurement errors, and human factors, thus providing a more comprehensive and realistic risk assessment. The fitted distribution of $NO_3^-$-N, $F^-$ and exposure variables are shown in Table S3.

## Statistical analysis

Monte Carlo simulations and sensitivity analyses were conducted using Crystal Ball software. Random forest analysis was performed in R language to identify key water environmental factors affecting water quality changes. Spatial distribution maps of various variables were plotted using ArcGIS 10.0.

## RESULTS AND DISCUSSION

### Hydrochemistry of groundwater

Statistical analysis of all hydrochemical variables of groundwater within the study area is summarized in Table 2. The pH values ranged widely from 3.05 to 11.09, with an average of 6.96, indicating the presence of acidic, neutral, and alkaline groundwater environments. TDS values varied between 25.38 and 1,635.21 mg/L, with an average of 213.56 mg/L. Mn concentrations ranged from below the detection limit to 19.93 mg/L, with an average of 1.45, and 38% of the samples exceeded the Class III groundwater quality standards as specified in the Groundwater Quality Standards (GB/T14848–2017). COD concentrations showed significant variation, ranging from below the detection limit to 8.57 mg/L, with 39% of the samples surpassing the Class III groundwater standards, and exhibited distinct spatial variability (coefficient of variation, CV = 140, Table 1). TH concentrations ranged from 9.47 to 905.78 mg/L, with an average concentration of 137.59 mg/L, and six samples exceeded the Chinese Class III groundwater standards. The most abundant cations and anions in the groundwater were $Ca^{2+}$ and $Cl^-$, respectively, with the dominance order of cations and anions being $Ca^{2+} > Mg^{2+} > Na^+ > K^+ > NH_4^+ > Fe_3^+ > Al_3^+$ and $HCO_3^- > SO_4^{2-} > Cl^- > NO_3^- > F^-$. Ammonium ($NH_4^+$) concentrations varied from 0.01 to 23.2 mg/L, with an average of 0.52 mg/L, and 15% of the samples exceeded the Class III groundwater quality standards (GB/T14848–2017). $Fe^{3+}$ concentrations showed significant variation among samples, ranging from 0.02 to 20.56 mg/L, with 27% of the samples exceeding the Chinese Class III groundwater environmental standards. $Al^{3+}$ concentrations ranged from 0 to 1.08 mg/L, with an average concentration of 0.07 mg/L,

Table 2 Descriptive statistics of groundwater hydrochemical parameters.

| Parameters | Min | Max | Mean | Std | CV (%) |
|---|---|---|---|---|---|
| $Na^+$ (mg/L) | 0.4 | 215.14 | 12.14 | 15.04 | 124 |
| $K^+$ (mg/L) | 0.02 | 48.9 | 5.63 | 7.08 | 126 |
| $Ca^{2+}$ (mg/L) | 0.1 | 285.11 | 34.68 | 28.04 | 81 |
| $Mg^{2+}$ (mg/L) | 0.53 | 69.55 | 12.38 | 9.4 | 76 |
| $Fe^{3+}$ (mg/L) | 0.02 | 20.56 | 0.4 | 1.23 | 308 |
| $Al^{3+}$ (mg/L) | 0 | 1.08 | 0.07 | 0.13 | 186 |
| $NH_4^+$ (mg/L) | 0.01 | 23.2 | 0.52 | 1.67 | 321 |
| $Cl^-$ (mg/L) | 0.62 | 748.84 | 19.72 | 42.06 | 213 |
| $SO_{42-}$ (mg/L) | 0.24 | 438.8 | 27.22 | 38.2 | 140 |
| $HCO_3$ (mg/L) | 6.03 | 639.05 | 137.61 | 100.87 | 73 |
| $F_-$ (mg/L) | 0.41 | 0.97 | 0.55 | 0.1 | 18 |
| $NO_{3-}$ (mg/L) | 0.23 | 73.89 | 7.26 | 8.99 | 124 |
| Mn (mg/L) | 0 | 19.93 | 0.49 | 1.45 | 296 |
| COD (mg/L) | 0 | 8.57 | 0.73 | 1.02 | 140 |
| TDS (mg/L) | 25.38 | 1,635.21 | 213.56 | 137.85 | 65 |
| TH (mg/L) | 9.47 | 905.78 | 137.59 | 99.68 | 72 |
| pH | 3.15 | 11.09 | 6.96 | 0.76 | 11 |

and 41 sampling points exceeded the Chinese Class III groundwater standards. It is evident that the Poyang Lake basin has exceeded Class III groundwater quality standards, potentially posing health risks that warrant attention. Moreover, the mean concentrations of groundwater chemistry in the Poyang Lake basin falls below both the World Health Organization standard and the levels in other regions of China and countries like Japan (Table S4), indicating that its groundwater pollution is relatively less severe.

Significant spatial differences in hydrochemical parameters were observed within the study area (Fig. S1). Sites exceeding the Class III groundwater standards for TDS and TH were primarily located in the upper reaches of the Ganjiang River, which may be associated with the scouring effect of higher flow velocities and resuspension induced by water conservancy construction (Li et al., 2016), and the typical strong lateritic and dry-cultivated soils in the basin, as well as extreme precipitation events, potentially exacerbating this phenomenon (Li et al., 2016; Pan, Cao & Zhang, 2020). Concurrently, the significantly higher concentrations of alkaline cations $Ca^{2+}$, $K^+$, $Mg^{2+}$, and $Na^+$ were also mainly concentrated in the Ganjiang River basin. The stronger rock weathering in the Ganjiang River Basin may lead to a substantial influx of cations into the groundwater. The enhanced evaporation during certain periods, such as the dry season, may also increase cation concentrations in groundwater through intensified rock weathering. Furthermore, human activities, such as the discharge of acidic wastewater, may promote rock weathering, resulting in an increase in cations like $Ca^{2+}$ (Li et al., 2017). The spatial variation in COD concentrations was significant (CV = 140), with sites exceeding the Class III groundwater standards primarily distributed in the Ganjiang and Xinjiang rivers. Both the Ganjiang and Xinjiang river basins contain typical agricultural development areas; the

upper reaches of the Xinjiang River are a typical agricultural zone, with effective irrigation areas accounting for 79% of the total basin area (*Li et al., 2021*; *Mao et al., 2023*), the use of fertilizers and pesticides in agricultural production may increase the organic matter content in the water bodies, leading to elevated COD concentrations. $F^-$, Mn, and $NH_4^+$ concentrations exceeding the Class III groundwater standards were found throughout the Poyang Lake Basin (Mn: CV = 296; $NH_4^+$: CV: 321). Additionally, $Al^{3+}$ and $Fe^{3+}$ concentrations showed distinct spatial differences (coefficient of variation, CV = 308), with high-concentration exceedance (Class III groundwater standards) sampling points mainly distributed in the study area except for the Xiushui River. Elements such as Fe and Al are ubiquitous in the watershed's rock strata and soils. During the dry season in the Poyang Lake watershed, factors such as geochemical weathering, reduced soil moisture, groundwater and table fluctuations may significantly contribute to the mobilization of these elements into groundwater (*Huang et al., 2022*; *Jiang et al., 2023*). These processes can enhance the release of Fe and Al through mineral dissolution and desorption, leading to their exceedance in groundwater. Regions with high $NO_3^-$ concentrations are primarily located in the middle and lower reaches of the Ganjiang and Raohe Rivers. These areas, characterized by flat terrain, are mainly devoted to agriculture and urban development. The use of agricultural fertilizers and the discharge of sewage and feces are likely causes of the elevated $NO_3^-$ concentrations (*Mao et al., 2023*). Within the study area, $NH_4^+$ and $NO_3^-$ show a significant negative correlation. They can interconvert: $NH_4^+$- oxidizes to $NO_3^-$ in oxic conditions, while $NO_3^-$ reduces to $NH_4^+$ in anaerobic settings (*Mao et al., 2021*). This suggests that spatial variations in redox environments may drive their inverse concentration relationship in this study. In the Poyang Lake basin, the water bodies with $F^-$ concentrations exceeding the Class III groundwater standard are mainly distributed in the upper reaches of the river. These areas host mining regions, such as the South China rare earth mine (*Cheng et al., 2024*). The weathering of these mineral resources and anthropogenic mining activities release substantial $F^-$, thereby influencing the $F^-$ levels in groundwater. Additionally, during the dry season, reduced groundwater recharge from precipitation and significant evaporation may also increase the regional $F^-$ concentrations (*Chen et al., 2013*).

The Piper diagram is extensively utilized for classifying groundwater hydrochemical types. In this study, within the Piper cation triangular plot, $Ca^{2+}$ is identified as the predominant cation, followed by $Mg^{2+}$; the anion triangular plot reveals $HCO_3^-$ as the primary anion, with $Cl^-$ as the secondary. Groundwater samples are predominantly clustered in Zone 1 of the triangular Piper diagram, followed by Zone 4, indicating that the predominant groundwater type in the study area is $HCO_3^-$-Ca, with a mixed Cl-Ca-Mg type as the secondary (Fig. 2). This classification aligns with the ionic concentration characteristics of the Poyang Lake Basin as discussed above.

## Water quality assessment
### Groundwater quality for drinking
The water quality of the study area is represented by calculating the groundwater EWQI (Fig. 3). The results show that the EWQI values of groundwater range from 15.11 to 945.26,

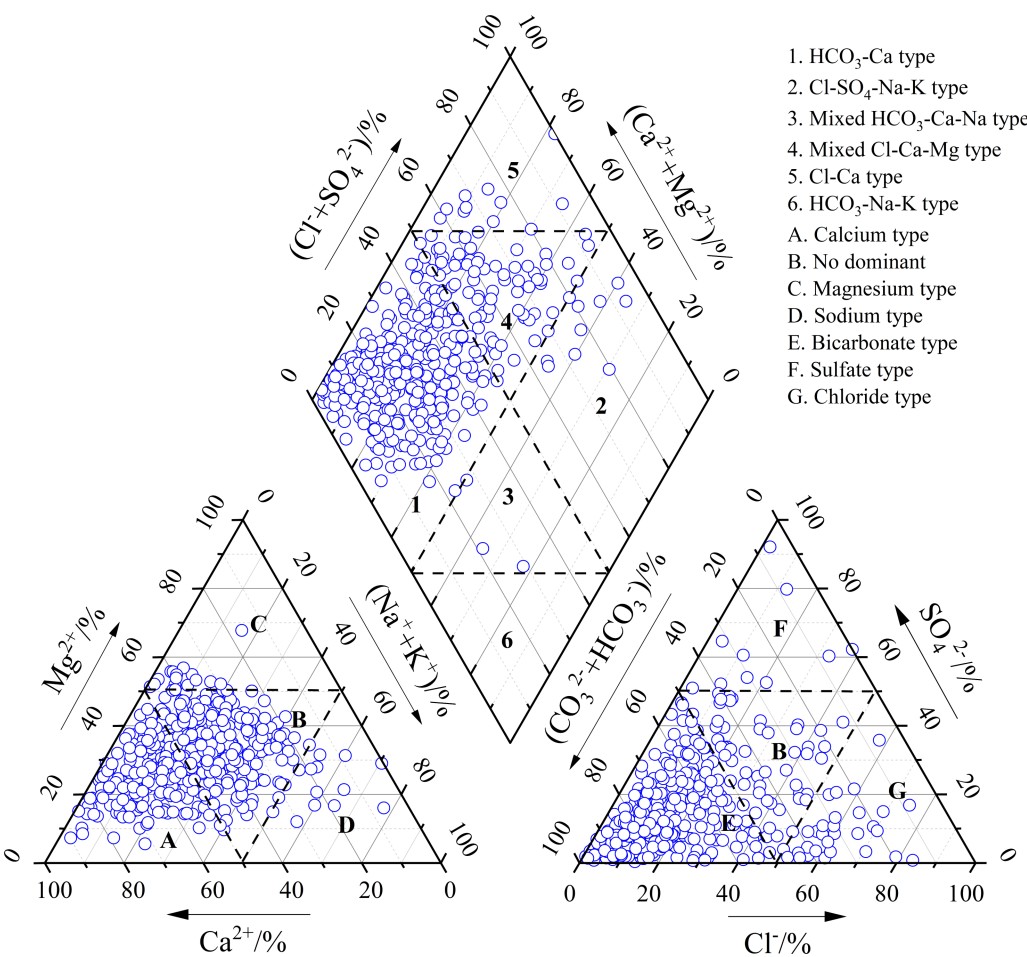

**Figure 2** Type of groundwater in study area.

with a mean value of 60.58, indicating that the overall water quality is acceptable, but there are areas where water quality has deteriorated. Within the study area, 26 sites have EWQI values between 100–150, and 30 sites have EWQI values greater than 150, indicating that the water quality at these sites has reached poor and very poor levels, respectively, with pollution being quite severe and not suitable for drinking water use. Additionally, polluted sites are widely distributed across various tributaries within the Poyang Lake basin (Fig. 4). Extensive human activities within the Poyang Lake basin, such as iron and coal mining and agricultural activities in the Ganjiang River basin, copper mining in the Rao River basin, large-scale agricultural activities in the upper reaches of the Xinjiang River, and widespread industrial production activities throughout the basin, may generate pollutants that, under the influence of abundant precipitation in the basin, can lead to erosion and infiltration (*Li et al., 2016*; *Li et al., 2021*; *Pan, Cao & Zhang, 2020*), thereby causing groundwater contamination.

We conducted random forest and sensitivity analyses on the EWQI and hydrochemical characteristics to determine which factors are more sensitive to EWQI (Fig. 4 and Fig. S2),

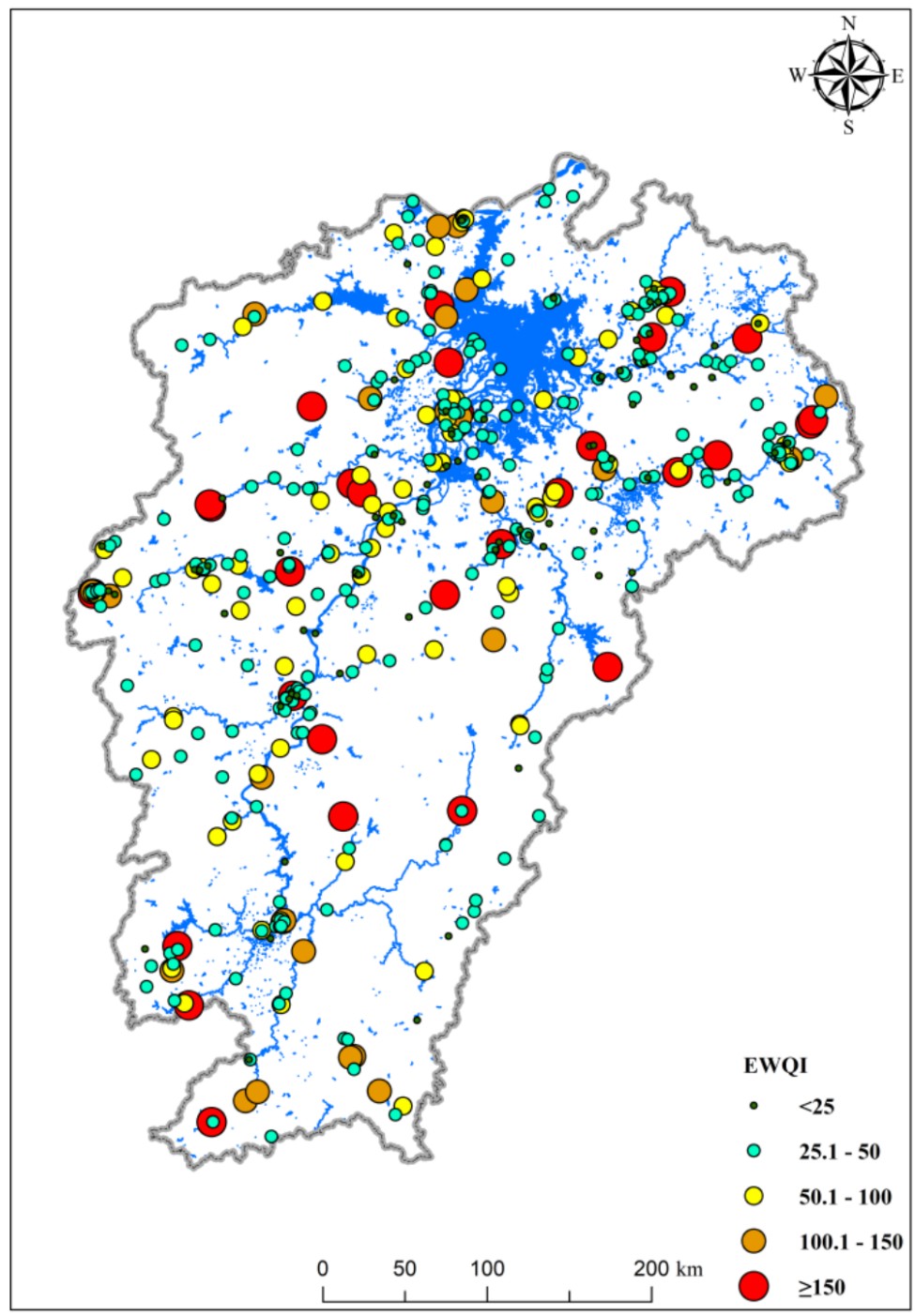

**Figure 3  EWQI values of study area.**

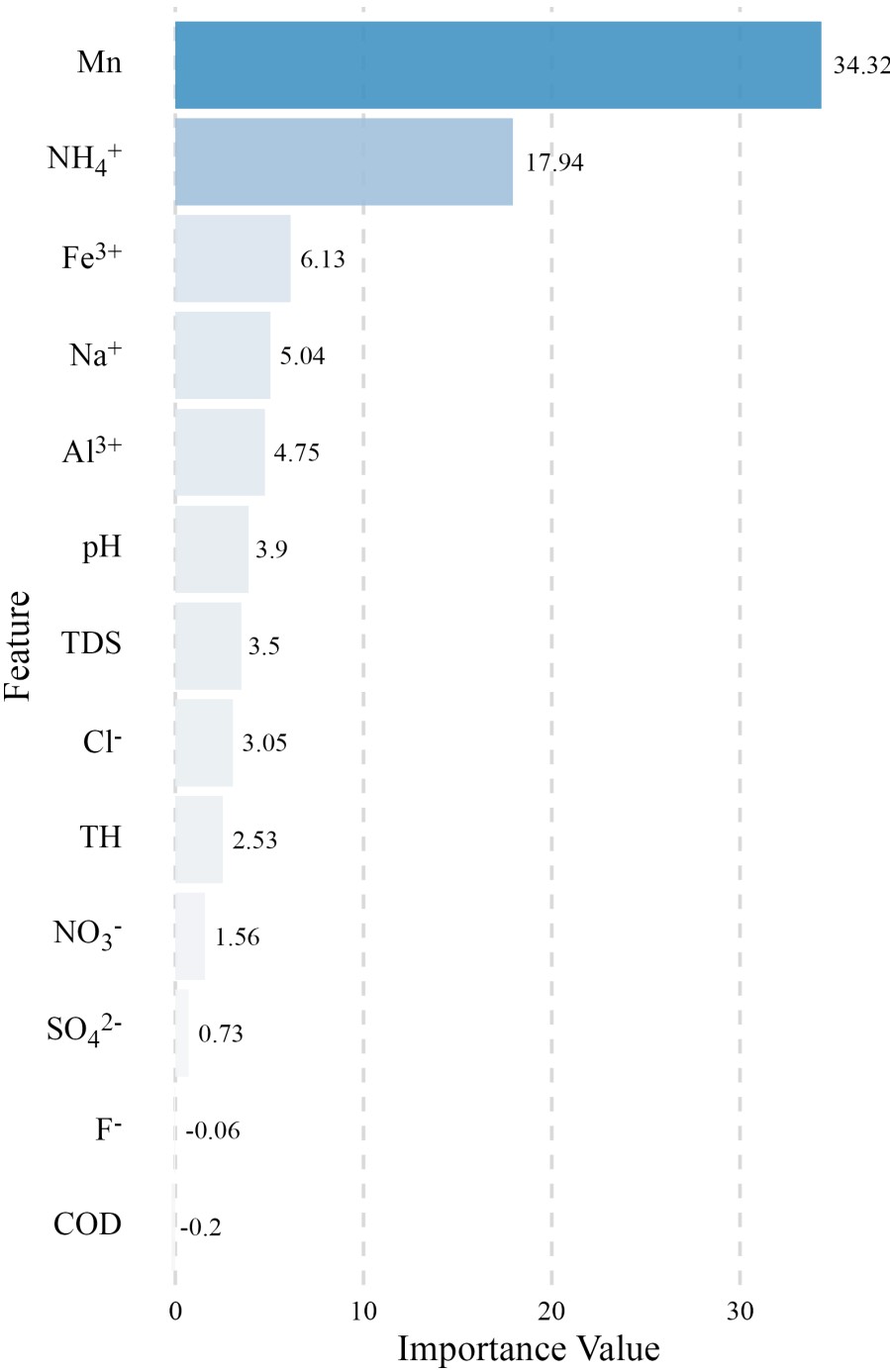

**Figure 4** **The importance of hydrochemistry on EWQI by random forest.**

thereby further identifying the sources of water pollution in the Poyang Lake basin. The random forest analysis results indicated that Mn and $NH_4^+$ are the primary influencing factors of EWQI, followed by Fe, Na, and Al. This is highly consistent with existing results regarding the surface water quality of the Poyang Lake basin, which considers

$NH_4^+$ as one of the most significant influencing factors of water pollution in the region (*Xu et al., 2022*). From the aforementioned analysis, it is evident that erosion processes, industrial wastewater, and non-point source pollution are the main causes of Mn and other exceedances in the basin, potentially being the primary controlling factors for water pollution within the basin. However, the sensitivity analysis revealed that EWQI is most sensitive to changes in Al and COD, suggesting that although they are not the primary contributors to water quality changes, even minor fluctuations can significantly impact water quality in the study area and warrant attention.

### Irrigation risk

Considering the extensive distribution of irrigation areas in the Poyang Lake basin, various irrigation risk indices such as SAR, KR, PS, PI, and MHR were employed to assess the quality of irrigation water, with the spatial distribution of these indices shown in Fig. 5. SAR serves as an indicator to determine the potential $Na^+$ accumulation (or sodium damage) in soil and plants due to irrigation water. Except for one site in the upper reaches of the Xinjiang River, SAR values in other areas are less than 18, indicating minimal sodium damage in the study area. In terms of KR, MHR, and PI, "unsuitable" irrigation sites are primarily distributed in the Ganjiang River and its estuary (KR > 1, MHR > 50, PI > 120), followed by individual sites in the Fu River, Xinjiang River, and Rao River (MHR > 50, KR > 1), suggesting that groundwater irrigation at these sites may cause alkaline damage. Furthermore, the soluble sodium percentage (SSP) evaluation results reveal widespread areas unsuitable for irrigation within the Poyang Lake basin (SSP > 60). Integrating these evaluation indices indicates that there may be extensive areas unsuitable for irrigation in the study area, with the Ganjiang River basin being more pronounced, which is consistent with the significantly higher concentrations of $Ca^{2+}$, $K^+$, $Mg^{2+}$, and $Na^+$ in the Ganjiang River basin (Fig. S1). Therefore, measures should be taken to prevent alkaline damage that may be caused by groundwater irrigation in the Ganjiang River Basin, especially in the context of frequent extreme dry conditions where groundwater resources become more critical for agricultural development.

## Health risk assessment
### Deterministic health risk assessment

Numerous studies have analyzed the significant health risks posed by $F^-$ and $NO_3^-$ in water bodies to human health (*Gao et al., 2022*; *Kaur, Rishi & Siddiqui, 2020*; *Padilla-Reyes et al., 2024*; *Shen et al., 2022*). This study employs the deterministic non-carcinogenic health risk assessment model recommended by the United States Environmental Protection Agency (USEPA) to evaluate the risks of $F^-$ and $NO_3^-$ in groundwater (USEPA, 2011). The model takes into account direct ingestion and dermal contact as exposure pathways, with relevant model parameters presented in Table S1. Figure S3 displays the results of the deterministic risk assessment. We found that different populations face varying risks. For children, the $NCH_{total}$ ranges from 0.64 to 3.37, with a mean of 1.04; for adults, the $NCH_{total}$ ranges from 0.27 to 1.45, with a mean of 0.44. Additionally, different contaminants have varying impacts on non-carcinogenic health risks to humans. For children, the $NCH_{F-}$ ranges from

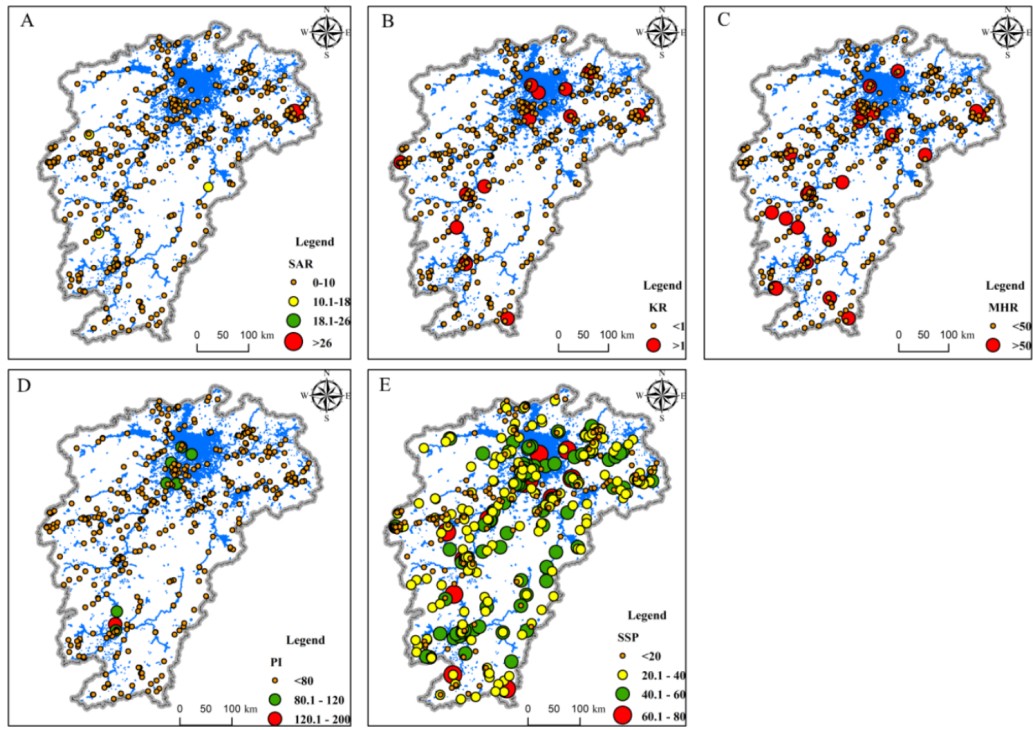

**Figure 5** **The irrigation risk assessment of groundwater.** (A) SAR; (B) KRl (C) MHR; (D) P1; (E) SSP.

0.58 to 1.37, with a mean of 0.78, which is significantly higher than for adults ($NCH_{F-}$, 0.24–0.59; mean: 0.34). Similarly, for children, the $NCH_{NO3-}$ ranges from 0.008 to 2.62, with a mean of 0.25, also higher than for adults ($NCH_{NO3-}$: 0.004–1.13; mean, 0.11). This indicates that groundwater in the Poyang Lake basin poses a significant non-carcinogenic health risk to humans, with a higher risk to children compared to adults, and $F^-$ poses a greater threat than $NO_3^-$. This is consistent with existing research, which suggests that children are more threatened by contact with groundwater, and $F^-$ poses a greater health risk than $NO_3^-$ to human health (*Kaur, Rishi & Siddiqui, 2020*; *Qiu et al., 2023b*; *Shen et al., 2022*; *Yang et al., 2022*).

Furthermore, the impact of various contaminants on human health varies. As shown in Fig. S3, 34 groundwater samples exceed the non-carcinogenic risk threshold for children's fluoride exposure (HI = 1), and no samples exceed the non-carcinogenic risk threshold for adults. For nitrate, there are 17 and 2 groundwater sites exceeding the acceptable risk levels for children and adults, respectively. This indicates that $F^-$ and $NO_3^-$ in groundwater pose non-carcinogenic health risks to humans only within certain regional scopes, with $F^-$ presenting a higher risk and broader impact, and the health risks posed by $F^-$ in the future should not be overlooked. Additionally, different exposure pathways have varying impacts on health risks. Figure S4 shows that the non-carcinogenic health risk from ingestion is much higher than that from dermal contact, a result consistent with the study by *Qiu et al. (2023a)*. Therefore, greater attention should be paid to the quality of drinking water, and

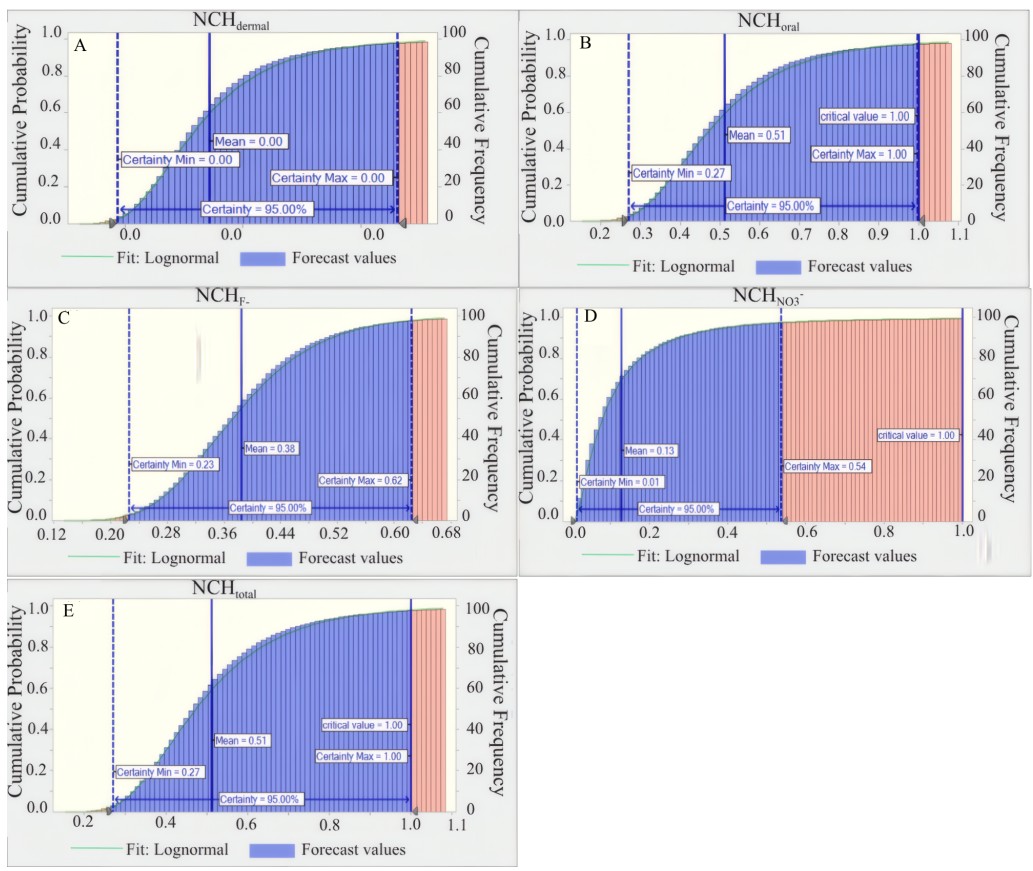

**Figure 6** **Probability risk assessment for children.** (A) NCH$_{dermal}$; (B) NCH$_{oral}$; (C) NCH$_{F-}$; (D) NCH$_{NO3-}$; (E) NCH$_{total}$.

ensuring the safety of drinking water by controlling the main exceedance contaminants in it is essential.

### *Uncertainty analysis*

Numerous studies have utilized Monte Carlo models to simulate health risks (*Gao et al., 2022*; *Padilla-Reyes et al., 2024*; *Qiu et al., 2023b*). Probabilistic modeling can clearly display the probabilities of various simulation outcomes and provide more comprehensive analysis results under conditions of limited data resources. This study employed the Monte Carlo method for probabilistic health risk assessment, with results shown in Fig. 6 and Fig. S4. The results indicate that the health risk probabilities for both populations generally follow a log-normal distribution. At the 95% confidence level, the range of NCH$_{total}$ for children is 0.27–1.10, with a mean of 0.51; for adults, the range is 0.08–0.29, with a mean of 0.15. The probability of children's NCH$_{total}$ exceeding the threshold (NCH = 1) is approximately 2%, while for adults, it is 0%. This suggests that the groundwater in the study area poses no health risk to adults, and although there is a certain health risk to children, the probability of risk occurrence is not high.

The health risk probabilities for different contaminants vary. At the 95% confidence level, the range of $NCH_{F-}$ for children is 0.23–0.62, with an average of 0.38; for adults, it is 0.06–0.18, with an average of 0.11. Concurrently, the range of $NCH_{NO3-}$ for children is 0.01–0.54, with an average of 0.13; for adults, it is 0–0.15, with an average of 0.04. Additionally, the probabilities of $NCH_{F-}$ and $NCH_{NO3-}$ exceeding the threshold ($NCH = 1$) for both children and adults are 0%. This indicates that the probabilities of groundwater contaminants $F^-$ and $NO_3^-$ posing non-carcinogenic health risks to humans are minimal.

The risk probability simulations for different exposure pathways of groundwater contaminants show that, at the 95% confidence level, the health risk range for children and adults through oral intake is 0.27-1 and 0.08–0.29, respectively, with average values of 0.51 and 0.15, respectively. The probabilities of health risks through oral intake exceeding the threshold ($NCH = 1$) for children and adults are approximately 2% and 1%, respectively. The health risks through dermal exposure are nearly 0. This demonstrates that, although the probabilities of non-carcinogenic health risks from different exposure pathways are relatively small, oral intake is the primary pathway through which groundwater in the Poyang Lake basin affects human health, with a greater impact on children's non-carcinogenic health risks. Previous studies also indicated the the higher health risk of water pollutants through oral intake, especially for children (*Yang et al., 2022*).

By comparison, it can be observed that the health risk assessment results for different contaminants obtained from deterministic analysis are to some extent confirmed in probabilistic simulation results. However, it is also evident that deterministic analysis overestimates the potential health risk impact of $F^-$ and $NO_3^-$ in groundwater to some extent. For instance, probabilistic simulation results show that the probabilities of $F^-$ and $NO_3^-$ posing non-carcinogenic health risks to humans are virtually 0, while deterministic analysis indicates that F- poses non-carcinogenic health risks at multiple sites across the basin. The overestimation by deterministic analysis may be attributed to its inability to account for the inherent variability and uncertainty in input parameters, which can lead to either over conservative or under conservative risk estimates (*Gao et al., 2022*; *Padilla-Reyes et al., 2024*). In contrast, probabilistic methods like Monte Carlo simulation provide a more nuanced understanding by incorporating the uncertainty distributions of contaminants' concentrations and exposure factors (*Asefa et al., 2024*). This nuanced approach is particularly valuable in regions like the Poyang Lake basin, where groundwater quality can be influenced by a multitude of variable factors (*Garg, Yadav & Aswal, 2019*), including agricultural practices, natural geological conditions, and hydrological dynamics. Therefore, our findings underscore the importance of integrating probabilistic methods into health risk assessment frameworks for groundwater contaminants. They offer decision-makers a more reliable basis for prioritizing risk mitigation strategies and allocating resources effectively. Moreover, the minimal risk probabilities for $F^-$ and $NO_3^-$ suggest that while these contaminants may not be the primary focus for stringent regulatory actions, continued monitoring and targeted interventions in high-risk areas (particularly those affecting children) are still warranted to ensure long-term public health protection.

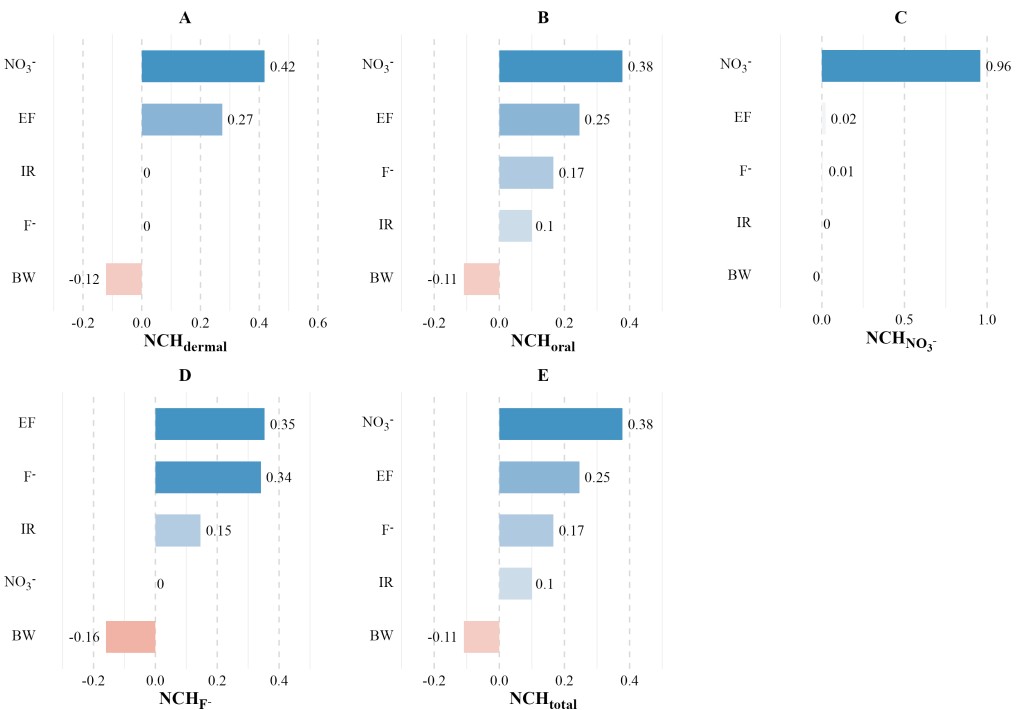

**Figure 7** **Sensitivity assessment for children.** (A) NCH$_{dermal}$; (B) NCH$_{oral}$; (C) NCH$_{NO3-}$; (D) NCH$_{F-}$; (E) NCH$_{total}$.

*Sensitivity analysis*

A sensitivity analysis was undertaken to evaluate the role of predictor variables in assessing non-carcinogenic health risks. The analysis was designed to pinpoint the key predictor variables that have a significant impact on the health risk assessment outcomes (Fig. S5 and Fig. 7). The sensitivity analysis indicates that for children and adults, the non-carcinogenic health risks from dermal and oral exposure to groundwater are sensitive to the factors in the order: $NO_3^- > EF > F- > BW > IR$. NCH$_{NO3-}$ is most sensitive to $NO_3^-$ concentration. For NCH$_{F-}$, the sensitivity order is EF/F- > BW > IR. NCH correlates positively with EF and pollutant concentration, and negatively with BW. High pollutant concentrations increase risk, and lighter individuals drinking contaminated groundwater frequently face greater non-carcinogenic health risks. These results align with *Gao et al. (2022)*, confirming that groundwater poses a higher non-carcinogenic health risk to children than adults. Therefore, controlling groundwater non-carcinogenic health risks should be tailored to local pollutant concentrations, with additional preventive measures for children, who are more vulnerable.

## Implication for water management and future research

The study's findings on groundwater quality in the Poyang Lake basin and the associated health risks highlight the need for targeted and region-specific water management strategies. There is a clear need to address the gaps in research, particularly regarding the specific toxicity parameters and exposure pathways relevant to the local population. The current reliance on USEPA guidelines for toxicity parameters such as AT and ED may not fully

apply to the local context, indicating a need for region-specific data. Future research should expand beyond direct ingestion to consider a broader range of exposure pathways, including dermal contact, and should also account for carcinogenic risks. Additionally, there is a critical need for longitudinal health outcome studies to establish a causal link between groundwater contamination and health effects, which can inform effective intervention strategies. Considering the potential impacts of climate change on water quality and the efficacy of management strategies is also essential. The socioeconomic implications of groundwater pollution and the costs associated with pollution control measures must also be assessed to understand their broader impacts on local communities and food security. In summary, the study calls for a comprehensive approach to water management that is grounded in localized research and addresses the complex interplay between environmental, health, and socioeconomic factors in the Poyang Lake Basin.

## CONCLUSION

This study analyzed groundwater quality in the Poyang Lake basin using 670 samples. The main conclusions of the study are as follows: (1) There were significant exceedances of Mn (38%), COD (39%), and $NH_4^+$ (15%) compared to China's Class III groundwater standards; (2) Piper diagram analysis indicated the dominant groundwater type is $HCO_3^-$-Ca, with a secondary Cl-Ca-Mg type; (3) While the overall water quality was acceptable based on the EWQI, 56 sites were identified as having poor to very poor quality, mainly due to Mn and $NH_4^+$; (4) Health risk assessment highlighted non-carcinogenic risks from $F^-$ and $NO_3^-$, particularly for children, though Monte Carlo simulations suggested these risks were minimal.

This study underscores the critical need for targeted water management strategies in the Poyang Lake basin to address groundwater pollution and protect public health. The findings provide a scientific basis for developing region-specific policies and interventions. Effective measures should focus on identifying and mitigating pollution sources, continuous groundwater monitoring, and improving water treatment infrastructure. These actions are essential to ensure sustainable water resource management and safeguard ecological and community well-being in the region.

### Funding

The manuscript is supported by Jiangxi Province Technology Innovation Guidance Program (2023KDG01008), Jiangxi Provincial Natural Science Foundation (20224BAB213036), National Natural Science Foundation of China (42261020); Jiangxi Provincial Key R&D Program for Young Scientists (20243BBH81035). The funders had no role in study design, data collection and analysis, decision to publish, or preparation of the manuscript.

### Grant Disclosures

The following grant information was disclosed by the authors:

Jiangxi Province Technology Innovation Guidance Program: 2023KDG01008.
Jiangxi Provincial Natural Science Foundation: 20224BAB213036.
National Natural Science Foundation of China: 42261020.
Jiangxi Provincial Key R&D Program for Young Scientists: 20243BBH81035.

## Competing Interests

The authors declare there are no competing interests.

## Author Contributions

- Xiaodong Chu conceived and designed the experiments, authored or reviewed drafts of the article, and approved the final draft.
- Jingyuan He performed the experiments, analyzed the data, authored or reviewed drafts of the article, and approved the final draft.
- Ting Chen analyzed the data, prepared figures and/or tables, and approved the final draft.
- Hailin You performed the experiments, prepared figures and/or tables, and approved the final draft.
- Xuhui Luo performed the experiments, prepared figures and/or tables, and approved the final draft.
- Shuping Liu performed the experiments, prepared figures and/or tables, and approved the final draft.
- Jinying Xu analyzed the data, prepared figures and/or tables, authored or reviewed drafts of the article, and approved the final draft.
- Zhifei Ma analyzed the data, authored or reviewed drafts of the article, and approved the final draft.

## Data Availability

Raw data is available in the Supplemental Files.

## Supplemental Information

Supplemental information for this article can be found online at http://dx.doi.org/10.7717/peerj.19937#supplemental-information.

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
