# Peer review of "Unveiling water quality and health risks from groundwater chemicals in Poyang Lake basin of China: a sophisticated analysis"

_PeerJ, doi:10.7717/peerj.19937_

## Round 0.1 · original submission · Major Revisions

Please refer to the reviewer comments.

Reviewer 1 ·

Basic reporting

This is routine research. It explains the water quality and health risk. I suggest the following suggestions to improve the manuscript.

1) The abstract section should highlight the aim and important findings of the research
2) Lines 53, 56, 141, and 149: The latest and relevant references may be cited
3) Explain how the current study differs from previous studies in this area.
4) Since these samples are from domestic wells and agricultural wells, they should be discussed separately to understand the variation in water quality between them.
5) ANOVA test between water quality of domestic wells and agricultural wells to confirm experimental results.
6) The conclusion section should highlight the important findings of the research.

Experimental design

None

Validity of the findings

None

Additional comments

None

Reviewer 2 ·

Basic reporting

The article is written in English and contains clear, unambiguous, technically correct text. The article complies with professional standards of courtesy and expression.

In the introduction, the authors only include references published by Chinese scientists that present the research results conducted in China, apart from two studies from India. Such a presentation of the problem is insufficient and completely ignores the broader field of knowledge.

The manuscript includes a results and discussion chapter. In my opinion, this part lacks a broader discussion of the results, a lack of confrontation with results from areas other than the Poyang Lake basin. There are no reports from other parts of China, Asia and the world. Although the authors try to explain the reasons for the presence of increased concentrations of individual chemical parameters, the arguments used are commonly known, e.g. the use of mineral fertilizers, sewage inflow, etc. There is a lack of specific information in this regard, e.g. where and which industrial plants causing water pollution are located, in what quantities mineral fertilizers are used in agricultural areas, etc. Some arguments are contradictory, e.g. higher evaporation - line 232-234, greater precipitation - line 249-251.

Experimental design

The authors have clearly defined the research questions that seem to be important and significant. The manuscript contains original data obtained by the authors during field and laboratory work. However, in my opinion, the research was poorly planned. Despite collecting 670 groundwater samples, no further measurement series were conducted. Based on one measurement series (1 sample), it is impossible to fully and accurately assess groundwater quality in the Poyang River Basin. For the assessment to be comprehensive, it is necessary to conduct further measurement series, e.g. in the wet season, in the following year. A full assessment of groundwater quality is also possible by conducting other analyses, e.g. heavy metals, etc., which were not conducted in these studies.

Validity of the findings

The data presented in the manuscript only partially present the qualitative state of groundwater. It is necessary to supplement the data with measurements from other periods of the year.

Additional comments

Other minor comments:

Fig. 1 lacks a broader background of the location of the study area in relation to the country, this part of the Asian continent

The characteristics of the study area include data on air temperature and precipitation that are not supported by any sources or are they the authors' own data?

In the description of the results, the authors use names such as the upper reaches of the Ganjiang River - line 226, the Ganjiang River Basin - line 231, etc. This river has not been described in Fig. 1, which significantly complicates interpretation. This comment applies to the other main rivers.

Reviewer 3 ·

Basic reporting

Introduction section needs more literature discussion to describe the research gap identified in the study area.

Revise the conclusion section and make it as point by point conclusion which must be related to study objective

Experimental design

Sampling and duration of sampling, period of sampling are missing
Study area description should be match with study objective
Detailed description of WQI and other calculation methods need to added in the methodology section.

Validity of the findings

In results and discussion, for each parameter section need to compare with previous studies.
Uncertainty risk study need discussion and compare the correlate value with source of contamination.

Additional comments

NA

---

## Round 0.2 · accepted · Accept

The authors have addressed all of the reviewers' comments.

Reviewer 1 ·

Basic reporting

The manuscript may be accepted for publication

Experimental design

The manuscript may be accepted for publication

Validity of the findings

The manuscript may be accepted for publication

Additional comments

The manuscript may be accepted for publication

Reviewer 3 ·

Basic reporting

Revised manuscript has been corrected in proper way

Experimental design

Revised manuscript has been corrected in proper way

Validity of the findings

Revised manuscript has been corrected in proper way